# Pretargeted Alpha Therapy of Disseminated Cancer Combining Click Chemistry and Astatine-211

**DOI:** 10.3390/ph16040595

**Published:** 2023-04-15

**Authors:** Chiara Timperanza, Holger Jensen, Tom Bäck, Sture Lindegren, Emma Aneheim

**Affiliations:** 1Department of Medical Radiation Sciences, Institute of Clinical Sciences, Sahlgrenska Academy, University of Gothenburg, 413 45 Gothenburg, Swedenemma.aneheim@radfys.gu.se (E.A.); 2PET and Cyclotron Unit, KF-3982, Copenhagen University Hospital, DK2100 Copenhagen, Denmark; 3Department of Oncology, Sahlgrenska University Hospital, Region Västra Götaland, 413 45 Gothenburg, Sweden

**Keywords:** astatine-211, targeted alpha therapy, pretargeting, click chemistry, radioimmunotherapy, radionuclide therapy, drug design, radiopharmaceuticals, theranostics

## Abstract

To enhance targeting efficacy in the radioimmunotherapy of disseminated cancer, several pretargeting strategies have been developed. In pretargeted radioimmunotherapy, the tumor is pretargeted with a modified monoclonal antibody that has an affinity for both tumor antigens and radiolabeled carriers. In this work, we aimed to synthesize and evaluate poly-L-lysine-based effector molecules for pretargeting applications based on the tetrazine and trans-cyclooctene reaction using ^211^At for targeted alpha therapy and ^125^I as a surrogate for the imaging radionuclides ^123, 124^I. Poly-L-lysine in two sizes was functionalized with a prosthetic group, for the attachment of both radiohalogens, and tetrazine, to allow binding to the trans-cyclooctene-modified pretargeting agent, maintaining the structural integrity of the polymer. Radiolabeling resulted in a radiochemical yield of over 80% for astatinated poly-L-lysines and a range of 66–91% for iodinated poly-L-lysines. High specific astatine activity was achieved without affecting the stability of the radiopharmaceutical or the binding between tetrazine and transcyclooctene. Two sizes of poly-L-lysine were evaluated, which displayed similar blood clearance profiles in a pilot in vivo study. This work is a first step toward creating a pretargeting system optimized for targeted alpha therapy with ^211^At.

## 1. Introduction

Targeted therapy involves an agent that selectively exerts a toxic effect on cancer cells and the tumor microenvironment to control cancer while minimizing harmful effects on healthy tissues. Targeted alpha therapy (TAT) is a type of cancer therapy in which radiation released from an alpha-emitting radionuclide is used as a means by which to produce cytotoxic effects on tumor cells [1]. The range of α-particles in tissues is about 25–100 µm, equivalent to just a few cell diameters, which is ideal for minimizing the irradiation of tumor-adjacent normal tissues [2], and their high energy of about 4–8.5 MeV makes them highly efficient at eradicating small clusters or isolated tumor cells [3]. TAT is typically delivered by attaching an alpha-emitting radionuclide to a biological molecule with targeting capability, such as monoclonal antibodies (mAbs), peptides, or antibody fragments. This vector selectively carries the alpha emitter directly to the target, resulting in a high radiation dose to the tumor cells with generally limited toxicity to the surrounding normal tissues [1,4]. The most common approach in targeted radionuclide therapy has been the use of mAbs. However, the excellent specificity of mAbs for their antigens is compromised by the exposure of healthy tissues to radiation during the long distribution time (24–48 h) required for the radioimmunoconjugate to circulate through the body and accumulate at tumor sites [5]. Therefore, when using mAbs as a targeting agent for cancer, the size of the antibody, i.e., its high molecular weight (about 150 kDa), generally results in slow tumor uptake, leading to a low tumor-to-normal tissue ratio [6]. Such a therapeutic strategy could be improved by combining short-lived alpha particle emitters with a smaller carrier molecule [7]. 

Pretargeted radioimmunotherapy (PRIT) is an interesting option that takes advantage of the specificity of mAbs to target the antigens expressed on tumors and at the same time it benefits from the fast binding and clearance of radiolabeled small molecules to allow for a high dose to the tumor while reducing the dose to normal tissue associated with directly radiolabeled mAbs [8]. In addition, the pharmacokinetic profile of PRIT allows the use of radionuclides with shorter half-lives in systemic settings that otherwise are not compatible with the long circulation time of full-size antibodies. In pretargeting, the treatment is given in two steps: (1) tumor targeting using mAbs, the pretargeting agent, and (2) dose delivery using a small vector, the effector molecule. Separating the slow targeting phase from the delivery of the radionuclide facilitates the optimization of the pharmacokinetics of radiation throughout the body. 

Today, several pretargeting strategies can be used to combine antibodies and radioligands in vivo; some of them utilize a noncovalent interaction such as streptavidin–biotin [9,10,11], and some utilize the hybridization of complementary oligonucleotides [12,13] or the ability of bispecific antibodies [14,15] to bind an antigen and a radiolabeled hapten. Last but not least, there is the biorthogonal inverse electron demand Diels–Alder (IEDDA) click reaction, which utilizes the fast and highly specific reaction that occurs between a diene, such as 1,2,4,5-tetrazine (Tz), and a dienophile, particularly trans-cyclooctene (TCO), which appears to be seven times more reactive than the cis-cyclooctene (CCO) in the IEDDA reaction [15,16,17,18,19]. The feature of biorthogonal chemistry is typically a click reaction that is feasible under mild conditions; it also has several advantages including that it is modular, wide in scope, stereospecific, and leads to inoffensive bioproducts. For clinical use, the products must be stable under physiological conditions and should ensure the biocompatibility of the reaction to avoid toxicity in living systems [20,21]. In this respect, the Tz–TCO pair is a good option, since neither of these is reactive to potential nucleophiles present in the biological system and their ligation leads only to the release of N_2_ (gas) as the only side product [22]. Therefore, the combination of selectivity, orthogonality, and rapidity of the IEDDA reaction makes it ideal for use in pretargeting applications. In PRIT, the antibody-based pretargeting agent is preferably functionalized by conjugation with TCO (TCO-mAb), while the effector molecule bearing the radioactive payload is functionalized with tetrazine (Tz-effector). 

Among the existing alpha-particle-emitting radionuclides, astatine-211 (^221^At), the second-heaviest halogen [23,24], offers many potential advantages for TAT. ^211^At decays with a half-life of 7.2 h in two ways, either through α-emission to ^207^Bi or electron capture (EC) to ^211^Po, which promptly decays through α-emission to stable ^207^Pb. The average range of the emitted α-particles in soft tissue is 57 µm and the average linear energy transfer (LET) is 97 keV/µm. Two important characteristics of ^211^At that differ from most other α-emitters relevant to TAT are that it yields one α-particle per decay [25,26], and also emits X-rays with energy between 77 and 92 keV during EC decay to ^211^Po. The polonium X-rays can be monitored by γ-cameras used in clinical diagnostic applications [3,27,28]. Iodine can, in many cases, be used as an analog to astatine for chemical optimization, and moreover, the iodine nuclides ^123^I and ^124^I are also relevant for diagnostic imaging applications [29,30,31]. Therefore, astatine and iodine could be used as a theranostic pair for combined therapy and imaging [32]. ^125^I is significantly more available than ^123^I or ^124^I and also displays decay characteristics such as low-energy gamma emission and a long half-life, which make it suitable for use in optimization [33]. 

Poly-L-lysine (PL) is a water-soluble polymer of the amino acid lysine that is commercially available in a large range of molecular weights. The key features of a radiolabeled effector molecule should be rapid and high accumulation at pretargeted tumor sites and fast clearance from the blood and normal tissues. An important factor affecting the pharmacokinetics of a molecule is its size [34]. Hence, the use of a polymeric scaffold will enable increased control in in vivo distribution, as polymers of different molecular weights can be employed. Moreover, PL allows for multiple functionalizations of its amino groups, e.g., different degrees of Tz groups can be attached to the backbone, and in this way, the avidity between the effector molecule (Tz-PL) and the pretargeting agent (TCO-mAb) can be improved. This potentially results in more effective pharmacokinetics with improved tumor targeting and increased absorbed dose to tumors [35]. 

The purpose of this study was to synthesize and evaluate a novel group of effector molecules for pretargeting applications with ^211^At and ^125^I, based on a PL scaffold, utilizing the Tz–TCO click reaction according to Figure 1. 

## 2. Results

The aim of this paper was to evaluate the produced effector molecule with regard to the effect of polymer chain length and functionalization, choice of radionuclide, and finally, binding properties and stability.

### 2.1. Poly-L-Lysine Functionalization

PL chain lengths of different sizes were investigated. There were two polymers with high molecular weight, referred to as the “large” polymers, one non-specific in the range of 15,000–30,000 g/mol (average 22,500 g/mol) and one specific at 21,000 g/mol. The low molecular weight polymer, referred to as the “small” polymer, was 10,000 g/mol. Their total chain lengths were assumed to be around 154, 143, and 68 lysine repeating units, respectively. Functionalization of the PL was conducted in three steps, according to Figure 1: with N-succinimidyl-3-trimethylstannyl-benzoate (m-MeATE) to enable radiohalogenation, with tetrazine-PEG5-NHS ester (H-Tz) or methyltetrazine-NHS ester (Me-Tz) for binding to TCO-functionalized mAb, and with succinic anhydride for charge modification.

When synthesizing the two (large and small) effector molecules, the aim was to functionalize between 5 and 15% of the lysines with m-Me-ATE and Tz groups in order to achieve good radiolabeling and efficient TCO binding while maintaining the structure of the polymer. The resulting lysine substitutions, depending on the functionalization yields for the different groups of the two polymer sizes, are shown in Table 1.

The PL functionalization ratios were determined using absorbance measurements, comparing the difference in absorbance of the modified PL to the calibration curve of Tz or m-Me-ATE. Trinitrobenzene Sulfonic Acid (TNBSA) analysis of free amine groups was also conducted; however, the experiments did not lead to reliable results. As PL is a homopolymer; when one of its amino groups has been trinitrophenylated, the neighboring lysyl residues become inaccessible, hence only a few of the amino groups can react with TNBSA because of the steric hindrance of the fixed trinitrophenyl group [36]. Electrospray mass spectrometry was also attempted to determine the functionalization ratios, but ionization of the highly charged polymer was found not to be possible. The structural integrity of the polymer after the functionalization steps was verified using Fast Protein Liquid Chromatography (FPLC) analysis (Figure 2). The retention times and curve shapes show that the functionalization of polymers with m-Me-ATE and Tz did not induce any fractionation or agglomeration of the two polymer sizes.

### 2.2. Radiolabeling

Three oxidizing agents, N-iodo-succinimide (NIS), N-chloro-succinimide (NCS), and chloramine-T (CAT), were evaluated in the radiohalogenation of large and small functionalized PL. For astatine labeling of functionalized PL of both molecular weights, NIS was found to be the most efficient; for iodine labeling, CAT performed best. It was also observed that, contrary to NCS and CAT, using lower amounts of NIS increased the radiochemical yield (RCY).

In addition to the different oxidizing agents that could be applied for radiohalogenation of the effector molecule, the effects of the concentration and amount of polymer were evaluated. It was found that the concentration of the polymer, no matter the molecular weight, did not impact the RCY if it was kept between 0.1 and 1 mg/mL. Similarly, the RCY could be maintained if the absolute amount of the functionalized polymer was kept above 0.05 mg. Applying lower amounts than this led to low yields and/or inconsistent results.

The best conditions for efficient and reproducible astatine labeling of conjugated PL of both polymer sizes were obtained by using NIS as an oxidizing agent (10 µL 20 µM for 5–25 MBq of ^211^At), which resulted in an RCY of 84.78% ± 0.03 for the small polymer and 85.38% ± 0.02 for the large polymer (based on >20 radiolabeling instances). For the iodination of both sizes of PL, the best conditions were obtained using CAT as an oxidizing agent (25 µL 8.79 mM for 1 MBq of ^125^I), in which the corresponding RCY values were 66.68% ± 0.14 for the small polymer and 91.16% ± 0.06 for the large polymer. The radiochemical purity (RCP) of both polymers was 97.7% ± 1.9 for the astatinated products and above 98% for the iodinated products. 

### 2.3. Specific Activity

We also attempted to increase the specific activity (SA) of the astatinated polymers by either increasing the starting activity or reducing the amount of polymer in the radiolabeling process. Low SA labelings usually yield 1 astatine atom in 10,000 polymers. More than 10 labelings with a SA higher than 1 astatine atom in 3000 polymers were performed yielding 3 times higher SA. The best results achieved so far with both methods are presented in Table 2. Lowering the amount of the effector molecule below 0.05 mg was found to be unsuccessful, as expected based on the results presented above. Increasing the starting activity resulted in a slightly reduced RCY especially for the small polymer, compared to the low-activity labeling. The SA can be expected to be further increased by starting with even higher initial activity. This was particularly evident for the large polymer, as high SA was achieved with a low polymer amount by applying only 57.6 MBq of astatine. In line with what is shown in Table 2, when trying to reach high SA the RCY varies depending on the starting activity and the amount and size of the effector molecule used in the labeling contrary to what happens in low SA labeling where the RCY is very consistent.

### 2.4. In Vitro Stability

After high-activity astatine labeling (starting activity: 124 MBq for 10 kDa and 110 MBq for 22.5 kDa), the stability of the effector molecules in 1 mL of physiological buffer, room-temperature phosphate-buffered saline (PBS), was investigated using repetitive measurements of the RCP over a period of 24 h for astatine. The radiolabeled effector molecules were also analyzed with FPLC right after labeling and 24 h later. The results, presented in Table 3, show that both sizes of PL displayed good stability by maintaining a high RCP. Figure 3 also shows that the structural integrity of both polymers was maintained after 24 h despite the solutions having received a dose of 4.5 and 4.0 kGy respectively, from astatine.

### 2.5. Pharmacokinetics

To evaluate the pharmacokinetics of the synthesized effector molecules, a pilot animal experiment was performed on four healthy female Balb C mice (two mice for each PL size), studying the blood activity profiles of iodinated effector molecules of both polymer sizes (large and small) after intravenous (IV) injection. One of the animals injected with the 10 kDa effector molecule received an extravascular injection and was therefore excluded from the study. As can be seen in Figure 4, both large and small polymer effectors quickly cleared from the blood, with most activity cleared within 5 h post IV injection and there was no apparent difference between the two polymer sizes. 

After the end of the experiment, organs were removed, and the results show that no accumulation of activity could be seen in the kidneys, see Appendix A. However, some retention could be found in the liver, which was more pronounced for the large polymer. This suggests that the excretion pathway of the effector molecules might be partly through the hepatic system, but this will have to be evaluated more thoroughly in future studies.

### 2.6. Effector Binding

To investigate the binding properties of the fully functionalized effector molecules, two Tz compounds were evaluated, as shown in Figure 1. H-Tz is known to have faster reaction kinetics to TCO than Me-Tz, however, it is also less stable. Two methods of binding were also evaluated: first TCO-modified beads, and when they were found to be successful, an actual pretargeting agent, a TCO-modified mAb, was employed.

#### 2.6.1. Effector Binding to Beads

As expected, the H-Tz effector was found to be faster in the reaction with TCO-functionalized agarose beads, leading to 40% binding within the first 5 min, and subsequently reaching >70% binding after 60 min. The Me-Tz effector showed ~30% binding after the first minute but did not reach >70% binding until after 24 h. Unspecific binding of the polymer to the filter membranes was also investigated, and it could be kept low (5.1%) with the binding performed in carbonate buffer at pH 8.5. Applying a buffer with physiological pH, PBS at pH 7.4, resulted in significant unspecific binding of 30%. This indicates only partial succinylation of the free amine groups of the PL, which at neutral pH can be in a protonated form, causing undesired reactions with the filter material. In addition, magnetic beads functionalized with TCO showed good binding of the effector molecules, but in this case, unspecific binding was even more prominent, and changing the buffer did not lower the unspecific binding enough to allow conclusions to be drawn from the results.

#### 2.6.2. Effector Binding to Pretargeting Agent

As the effector binding to beads showed promising results, a pretargeting agent was synthesized by reacting TCO-NHS with lysine groups on the mAb trastuzumab (Herceptin^®^). The binding between the pretargeting agent and the effector molecule was evaluated in vitro using two methods: precipitation and FPLC. In the precipitation method, only the antibody precipitates while the unreacted polymer remains in solution (Figure 5). By radiolabeling the effector, the amount of the effector that has reacted with the pretargeting agent can be evaluated by precipitating the click product. Using the large PL (21 kDa), the results show that for the H-TzPL, the reaction was already complete within the first 5 min (all effector activity in precipitate), while the Me-TzPL reacted completely in 15 min (97% of effector activity in precipitate). When using the small PL (10 kDa), the reaction kinetics was found to be slower with 92% of effector activity in the precipitate for the HTzPL and 52% for the MeTzPL after 5 min. The reaction was completed in 15 min for HTz-substituted PL and in 60 min for the MeTz. A slower reaction kinetics might be due to the lower Tz-functionalization of the small PL. FPLC analysis of the click products was performed to confirm the reaction between the labeled TzPL effector and the TCO-mAb pretargeting agent. FPLC fractions were collected, and the activity was measured on a γ-counter. The results, represented in Figure 6, show that the peak retention time was shorter for the clicked product (large PL-mAb 17.5 min ± 0.7; small PL-mAb 23.3 min ± 0.6) than the TzPL (large PL 28.9 min ± 0.2; small PL 30.5 min ± 0.6), indicating a larger molecule. This retention time is also shorter than that analyzed by UV for TCO-mAb (Rt = 28.5 min ± 0.2).

## 3. Discussion

The focus of this study was developing and evaluating a novel PRIT strategy based on a click chemistry approach, employing a PL-based scaffold as the effector molecule. As different sizes of polymers are known to have different circulation times in vivo, two sizes of PL, one large and one small, were investigated to ultimately optimize the pharmacokinetics of the effector molecule. For the large PL, a high-molecular-weight polymer, in the range of 15–30 kDa, was initially used to allow high multi-functionalization. Subsequently, PL of fixed molecular weight (21 kDa) was employed, applying the same conjugation conditions. As different PL sizes can affect the pharmacokinetics, different PL sizes will also allow different degrees of functionalization, which in turn, will affect the specific activity (activated tin esters, ATE) and the avidity (Tz) of the effector molecule towards the pretargeting agent. The polymer was conjugated with the different agents and high reaction yields were achieved (92% for m-Me-ATE and 57.4% for Tz). However, when the same protocol and ratios were applied to a smaller polymer, precipitation occurred, and adjustments had to be made in order to achieve successful synthesis. The functionalization ratio was decreased for both m-MeATE and Tz, which enabled complete synthesis without precipitation at any stage and high reaction yields (52% for m-Me-ATE and 58% for Tz). The reason for the precipitation of the low-molecular-weight PL was likely overloading of the polymer with reagents, resulting in higher hydrophobicity compared to the high-molecular-weight PL. In both cases, a sufficient number of lysine groups were functionalized.

Radiolabeling of all four types of effector molecules was shown to be successful. In both astatination and iodination of the polymers, high radiochemical yields with high radiochemical purity were achieved. Astatination yields (85.03% ± 0.03) were higher compared to the RCY generally achieved in the astatination of antibodies utilizing the same prosthetic group and oxidizing agent [33,34]. When a low amount of effector was used, it was observed that the RCY varied according to both the type and amount of oxidizing agent employed in the labeling. The amount of NIS had to be at least 20 times lower than NCS or CAT to reach a high RCY. This is likely due to a competing destannylation reaction between tri-iodine ions liberated from NIS and the radionuclides, leading to a lower RCY. The same competing reaction will not occur when using a chlorinated oxidizing agent, e.g., NCS or CAT [37]. The effect of the degree of functionalization of the polymers was also evident in the RCY. Upon iodination, it was significantly higher for the larger, more substituted polymer compared to the small one. 

After labeling, the SA of the effectors was assessed. Decreasing the effector amount below a certain level was found to be unsuccessful, most likely due to reaction conditions, including radiolysis, occurring in the highly diluted solutions [38,39]. However, high SA could be obtained by reducing the effector amount to 0.05 mg and increasing the starting activity. This suggests that the SA of the effectors could be further enhanced compared to the data presented here by increasing the starting activity applied during radiolabeling while maintaining a low effector concentration. It was found to be easier to achieve higher SA for ^211^At using the large effector molecule and applying lower activity, compared to the small effector. This was expected due to the larger presence of prosthetic groups for radiolabeling. Putting the concept of SA into a clinical context, we could apply a starting activity of around 1 GBq, which, assuming the RCY is maintained for the labeling reactions as presented in Table 2, would render an SA value in the region of 1:100 (astatine:polymer). With that said, the SA of both polymer sizes in this work is already clinically relevant when compared to, e.g., directly radiolabeled antibodies [2]. So far, only two clinical studies have been conducted with ^211^At. In both studies, an activity of around 300 MBq was used as the final dose escalation [40,41]. With the SA levels presented in this paper, this would render a polymer mass of 0.1–0.3 mg, which could be considered safe for patient administration. 

The synthesized effector molecules showed good stability for up to 24 h, which should be more than enough, given the fast blood clearance, with most activity having left the blood system after 5 h. The circulation time of the effector molecules is also well in line with the 7.2 h half-life of astatine. 

The binding properties of the effector molecule against a TCO-modified pretargeting agent showed rapid binding at physiological pH; after only 5 min, over 90% of both effectors had already reacted when using the large PL while the small PL showed slightly slower reaction kinetics. The good binding properties of the pretargeting agent in combination with the blood profile indicate great promise for efficient in vivo application.

In this study, it was shown that by utilizing the Tz-TCO click chemistry system, several different PL effector molecules for pretargeting can be synthesized and successfully functionalized for efficient radiolabeling and binding without compromising the structural identity of the polymer. It was shown that it is possible to stably label the effector with ^125^I, as a surrogate for imaging nuclides ^123^I and ^124^I, and the rare therapeutic alpha-particle emitter ^211^At with high RCY, RCP, and SA, while still maintaining good binding to TCO-functionalized mAbs. In future studies, these promising new effector molecules will be further assessed in in vivo models with respect to, e.g., biodistribution and tumor targeting with TCO-modified mAbs.

## 4. Materials and Methods

### 4.1. General

Commercial [^125^I]-NaI was obtained from PerkinElmer in 10^−5^ M NaOH solution with an activity concentration of >350 mCi mL^−1^. ^211^At was obtained from the PET and Cyclotron Unit at Copenhagen University Hospital (Denmark). The nuclide was transformed into a chemically useful form by dry distillation at the Sahlgrenska Academy (Gothenburg, Sweden) [42]. m-MeATE bifunctional labeling reagent with 97% purity was purchased from Toronto Research Chemicals, Inc. Toronto (ON), Canada; 10,000 and 21,000 Da poly-L-lysine hydrobromide were purchased from Alamanda Polymers Inc. Huntsville (AL), USA; TCO-NHS ester was purchased from Click Chemistry Tools, Scottsdale (AZ), USA; and all other chemicals included in this study were obtained from Sigma Aldrich, Inc. St. Louis (MO), USA, and were of at least analytical grade.

### 4.2. Polymer Conjugation with N-Succinimidyl-3-(Trimethylstannyl)-Benzoate and Tetrazine-NHS Ester

In this study, PL was used as an effector molecule scaffold. The polymer was conjugated in advance before radiolabeling in three steps. PL was dissolved in 0.2 M carbonate buffer (pH 8.5) at a concentration of 4 mg/mL. From a 50 mg/mL stock solution of m-MeATE in chloroform, an aliquot was transferred to a glass micro vial (1.1 mL V-vial, VWR), and the solvent evaporated. The residue was dissolved in dimethyl sulfoxide (DMSO), resulting in a concentration of 115 nM. The m-MeATE was added to the PL solution and allowed to react at room temperature (RT) under gentle agitation for 30 min. Then, H-/Me-Tz was dissolved in DMSO to a concentration of 200 mg/mL and added to the reaction mixture, and the reaction proceeded for another 30 min. Solid succinic anhydride was then added in four times molar excess. The pH was adjusted with 20 µL of 1 M sodium carbonate to maintain a level of around 8.5. After 30 min, the resulting conjugated polymer was purified by size exclusion chromatography using an Amersham NAP-10 column Cytiva–GE Healthcare, Buckinghamshire (UK). The product was eluted in a PBS solution at pH 7.4. Table 4 shows the corresponding PL and reagent amounts.

### 4.3. Electrophilic Astatination

Upon labeling, the buffer of the conjugated polymer was exchanged from PBS to 0.2 M acetate, pH 5.5, to enable the astatine labeling reaction. Astatination was performed starting with 2–200 MBq of ^211^At in chloroform obtained from the irradiated target using an Atley C100 module, Atley Solutions AB (Sweden). The solvent was evaporated under a gentle nitrogen stream to obtain a dry residue of ^211^At, to which N-iodosuccinimide (NIS) (10 µL; 20 µM) in methanol (MeOH)/1% acetic acid (HAc) was added to oxidize the radionuclide. After 30 s, conjugated PL (large/small PL: 100 µL, 1 mg/mL) in acetate buffer, pH 5.5, was added to the reaction vial. The labeling of the polymer proceeded under agitation for 1 min, after which NIS in MeOH/1% HAc (3.1 µL, 22.2 mM) was added to the reaction mixture to exchange the remaining tin groups for iodine. After 1 minute of agitation, the reaction was quenched with ascorbic acid in water (5 µL, 0.27 M). The polymer product was isolated in PBS using aNAP-10 column Cytiva–GE Healthcare, Buckinghamshire (UK).

### 4.4. Electrophilic Iodination

Iodination of the polymer was performed essentially using the same method as for astatination, except for a few different conditions. A few microliters of ^125^I in 10^–5^ M NaOH were diluted with 0.01 mM NaOH to a concentration of 1 MBq/µL. Unless otherwise stated, CAT (25 µL, 2 mg/mL) was used as an oxidizing agent to activate the ^125^I (5 MBq in 0.01 mM NaOH). The reaction was stopped by the addition of Na_2_S_2_O_5_ in water (21 µL, 2 mg/mL). The polymer product fraction was isolated in PBS by passage over a gel filtration NAP-5 column.

### 4.5. Radiochemical Purity

The RCP of the iodinated PL was assessed using thin-layer chromatography (TLC) on TLC Silica gel 60 F₂₅₄ 25 Aluminium sheets, Merck, Darmstadt (Germany), in 80% MeOH/ethyl acetate (EtOAc). The TLC strips were cut into three pieces for analysis and counted for radioactive content in a NaI(Tl) γ-counter Wizard 2480, PerkinElmer (Singapore). While free ^125^I migrated to the front, the labeled polymer stayed at the baseline, enabling good separation.

The radiochemical purity of the astatinated PL was measured on a miniGITA Dual radio TLC scanner with an alpha probe, Elysia-raytest Straubenhardt (Germany) using iTLC-SG glass microfiber chromatography paper impregnated with silica gel, Agilent Technologies, Folsom CA, (USA). After a TLC run in ethanol of the astatinated effector molecule [43], the strip was placed on the radio TLC scanner. In these conditions, free astatine migrates to the frontline while the effector molecule stays in the deposit spot, as shown in Figure 7. 

### 4.6. Analysis of Polymer Composition

The polymer composition was evaluated using a SpectraMax QuickDrop UV-Vis spectrophotometer, Molecular Devices, San Jose, CA (USA) in which the detected difference in absorbance was used to determine the functionalization degree of the polymer with respect to Tz and m-Me-ATE after comparing their respective calibration curves. The calibration curves were constructed by measuring in triplicate 10 µL of different concentrations of m-Me-ATE in DMSO/water at 250 nm and Tz in PBS at 270 nm. Then, the absorbance of the functionalized polymer at the two wavelengths was measured and subtracted from the absorbance of the naked reference PL. From the value of the difference in absorbance and the calibration curve, it was possible to identify the concentration of the functionalized groups attached to the polymer.

### 4.7. FPLC Analysis of Conjugated Polymer

The structural integrity of the polymer was evaluated on an FPLC system ÄKTA purifier, Amersham Biosciences, GE Healthcare, (Sweden) with a Superdex 200 10/300 GL column Cytiva-GE Healthcare Bio-Sciences AB, Uppsala (Sweden) using a flow rate of 0.5 mL/min, comparing the UV chromatograms (225–280 nm) obtained from the succinylated and functionalized PL with the radioactive chromatogram obtained by fraction collection, 1/min, of the astatinated PL, measured using a NaI(Tl) detector.

### 4.8. Stability Test 

The stability of the labeled effector molecule was evaluated by analyzing the radiochemical purity (see Section 4.5) and by following the activity distribution after FPLC fraction collection (see Section 4.7) over a period of 24 h.

### 4.9. Animal Study 

The blood distribution of the two sizes of iodinated effector molecules was evaluated in healthy female normal Balb/c mice. Four mice were divided into two groups, two mice for each PL size (A and B, 21 kDa; C and D, 10 kDa). Following IV injection of the effector molecule (100–140 kBq in 100 µL of PBS), blood samples were taken after 30 min, 45 min, 2 h, 4 h, and 20 h and measured on the γ-counter. Mice were sacrificed by cervical dislocation after 24 h. Whole blood was collected by cardiac puncture immediately after the animals were killed, and tissues including salivary gland, throat (including thyroid), lungs, heart, stomach, liver, spleen, small intestine, large intestine, kidneys, intraperitoneal fat, femur, and muscle were removed. The tissues were accurately weighed and measured on the γ-counter. One animal, D, received an extravascular injection and was therefore excluded from the study. 

### 4.10. Tz-TCO Binding to Beads

In order to evaluate the binding of the effector molecule, agarose beads functionalized with TCO were used for reaction with Tz-conjugated PL. TCO-agarose beads were obtained from Click Chemistry Tools, Scottsdale AZ, (USA) in a concentration of 10–20 µmol TCO groups per mL of resin. Costar Spin-X centrifuge tube filter, 0.45 µm from Corning Incorporated, Salt Lake City UT (USA), was used to assess the binding percentage, separating the unbound fraction of TzPL from the one bound to the TCO beads. After assembling the centrifuge tubes, the filter was treated with 500 µL bovine serum albumin (BSA)-azide in PBS and then centrifuged to minimize unspecific binding. TCO-agarose beads (20 µL, 15 mM) were added to the filter and centrifuged in an Eppendorf centrifuge 5702, Hamburg (Germany)for 5 min at 4400 rpm. The beads were then washed with carbonate at pH 8.5 and centrifuged again. Then, labeled TzPL eluted in carbonate at pH 8.5 (217 µL, 1.4 mM) was added to the tube. Unless otherwise stated, the labeled TzPL was allowed to react with TCO beads for 1 h. At the end of the reaction, the mixture was centrifuged for 5 min at 4400 rpm, washed with carbonate at pH 8.5, and centrifuged again. The reservoir (filter) was separated from the centrifuge tube containing the unbound filtered fraction and the washing buffer and measured separately on the γ-counter. The binding yield was calculated by dividing the activity in the reservoir by the sum of the activity in the reservoir and the centrifuge filter. Unspecific binding was evaluated on the filter of the centrifuge tube without TCO-agarose beads, measuring the activity of the polymer left in the filter after centrifuging and washing.

### 4.11. Tz-TCO Binding to Magnetic Beads

MagnaBind Amine Derivatized Beads (amine concentration = 12 µmol/mL) from Thermo Scientific, Rockford IL (USA) were also used to assess the binding of the PL effector after functionalization with the TCO-NHS ester (Click Chemistry Tools). For this procedure, 103 µL of magnetic beads (mBeads) were taken from the stock and washed three times with 1 mL of carbonate buffer at pH 8.5 before adding 65.4 µL of TCO previously dissolved in DMF at a concentration of 25.4 mg/mL. The mBeads were left to stir for 30 min protected from light to preserve the transconfiguration of the TCO for the reaction with Tz [19]. After half an hour, the mBeads were washed three times with carbonate and dissolved in 1.2 mL of carbonate, and 200 µL of 211At-HTzPL and 211At-MeTzPL were added to two reaction vials, each containing 200 µL of TCO-mBeads, corresponding to 10 times excess TCO over Tz, to evaluate the binding. A sample of 50 µL was taken out of the vial and measured on the γ-counter, and later it was washed with PBS and measured again to determine the binding. Unspecific binding to the TCO-mBeads was evaluated by measuring the activity left on the mBeads after reacting with polymer not bearing Tz groups.

### 4.12. Synthesis of Pretargeting Agent

The pretargeting agent was synthesized by conjugating TCO-NHS to the mAb trastuzumab (Herceptin^®^). The antibody was purified twice, first on a NAP-5 then on a NAP-10 column, and eluted in 0.2 M carbonate buffer at pH 8.5 to a concentration of 5 mg/mL. TCO-NHS was dissolved in DMF to a concentration of 10 mg/mL. Then, 2.2 µL of TCO-NHS was added to 500 µL of the mAb solution, and the mixture was left to stir protected from light for 2 h at RT. When the reaction was complete, the conjugated antibody was purified on a NAP-5 column and the product was eluted in PBS.

### 4.13. Tz-TCO Binding Precipitation

In this study, the precipitation of an antibody in the presence of an excess of methanol, was used as a method to evaluate the binding efficiency between an effector molecule and a pretargeting agent. The conditions mimicked the clinical setting; therefore, after radiolabeling, 0.16 nmol, 3.3 µL of labeled effector molecule was allowed to react with 1.6 nmol 500 µL of pretargeting agent in an Eppendorf tube for 1 h. The binding was tested in triplicate by adding to each tube 200 µL of BSA in PBS, around 1 kBq of the reaction mixture, and 500 µL of methanol to obtain a white precipitate/cloudy solution. The three tubes were then measured on the γ-counter and centrifuged for 5 min at 4400 rpm. The supernatant was separated from the pellets by water suction, and the remaining activity was measured again on the γ-counter to evaluate the amount of PL that had clicked to the TCO-mAb. 

## Data Availability

Data is contained within the article and Appendix A.

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
