# Peer review of "Pretargeted Alpha Therapy of Disseminated Cancer Combining Click Chemistry and Astatine-211"

_pharmaceuticals, 2023, doi:10.3390/ph16040595_

Round 1
Reviewer 1 Report (Previous Reviewer 1)
The revised manuscript showed that the authors performed the requested corrections and amendments.
There are still the following minor issues to address:
11) The major tick marks are missing from the x-axis of all figures. It is confusing and aesthetically unpleasant.
22) The results shown in table 2 and figures 4, 5 and 6 do not report any SD values or uncertainty bars giving the impression that the authors performed the experiments only once (and n = 1 is a poor statistical power). Please amend accordingly.
Author Response
We have now provided a point-by-point response to reviewer #1´s comments, please see the attachment. We thank both reviewers for taking the time to review our manuscript and for their suggestions to improve its form.

Reviewer 2 Report (Previous Reviewer 3)
I thank the authors for addressing my minor concerns and for general improvement in the manuscript
Author Response
Reviewer #2 supported the manuscript and had no comments regarding its present form. We thank both reviewers for taking the time to review our manuscript and for their suggestions to improve its form.
This manuscript is a resubmission of an earlier submission. The following is a list of the peer review reports and author responses from that submission.
Round 1
Reviewer 1 Report
The manuscript entitled “Development of 211At and 125I Radiopharmaceuticals for Pretargeted Radioimmunotherapy of Disseminated Cancer” describes the preparation and assessment of polylysine-based radioconjugates for IEDDA.
Sadly, the manuscript has many serious issues, and it cannot be considered for publication in Pharmaceuticals.
11) The work shown in the manuscript has limited novelty and impact. The described experiments should be implemented with complete radioiodination optimization and some in vitro and in vivo data to be considered for publication. With the current structure, even after major revisions, the manuscript is not suitable for “Pharmaceuticals” and should be submitted to an alternative journal (lower IF).
22) The whole manuscript displays poor English (spelling mistakes, terms used wrongly, spurious symbols) and multiple long and confusing sentences. Even the references style is all over the place. Sadly, the fluidity of the manuscript is affected by all the above.
33) Some crucial pieces of information are missing and there are a lot of imprecisions that should be taken care of.
Some examples:
Introduction:
The following points should be elaborated a bit more:
The authors use polylysine polymers but there are no details on the molecule itself, or why it is used. Please add few lines about the subject.
In page 2, the authors mention iodine-125 but it is not clear why they use iodine and why they use that specific radioisotope. The authors should include a more detailed and informative paragraph on the subject.
The aim of the work shown in the manuscript was more or less clear, but the final aim is not very clear (suddenly and very shortly a TCO-mAb appears in the text): the authors should indicate in more detail what they plan to use the effector for.
Results:
Page 3: The description of the numbers listed in Table 1 is chaotic and uninformative. The authors should restructure it and describe the Table clearly, focusing on the most relevant ratios and characterisation (analytical methodologies that did not work should be discussed in Discussion, not described in Results)
Separate schemes showing the radiolabeling reaction and the IEDDA reaction and products would be useful.
Page 5, section 2.3.: the last four sentences of the section describe (not very clearly) a product that is not included in Table 3. Additionally, it is not clear what are the best conditions for an efficient and reproducible radiolabeling of each polylysine conjugate. Please indicate.
Discussion:
Few lines explaining why the authors use polylysine polymers of different length should be added in Discussion. The reason might be lost and be unclear. The authors should also explain why they aim to bind so many Tz and Sn functionalities on the same molecule. It might not be obvious.
Page 7: The paragraph regarding the type of buffer and the influence on the unspecific binding should be in Results and not in Discussion.
Materials and Methods:
Is the term Tetrazine-N-hydroxysuccinimide ester a general term for both tetrazines (H-Tz and Me-Tz)? It is ambiguous.
Are the procedures described in section 4.2., 4.3. and 4.4. the optimized conditions for conjugation, astatination and radioiodination, respectively? What about the optimization procedures (e.g. use of different amounts of precursors, different oxidising agent, etc.)? They are missing.
The details about the purification of the radiolabeled compounds are missing (e.g. type of column and eluent, etc.). Please add.
Section 4.5. is full of unnecessary details. Please simplify and focus on the most important information. This section might even be superfluous and can be added as QC to the relevant sections (4.3 and 4.4).
Section 4.7.: what column and mobile phase were used?
The procedure described in section 4.8. do not include all the time points indicated in Figure 3. Please correct accordingly.
44) Tables and Figures:
The chemical structures in Figure 1 (page 3) are barely visible and the whole reaction scheme (it should be Scheme1, not Figure1) is quite confusing. The structure shown is that of a tri-lysine not a polylysine. The authors forgot the parentheses. Additionally, it looks like the authors added all the reagents in the same reaction mixture and got two products. The reaction scheme should be changed and clearer.
Table 1 (page 4): the table should be properly explained in the text. It is not clear what “theoretical yield” and “actual yield” are.
Figure 2: The unit on the Y-axis is missing. FPLC is the only QC method shown by the authors for the characterization of the conjugates and is it barely described in Results. The authors should include more details. Additionally, the radioactive chromatogram is included in the figure cluttering it. The quite crowded figure should be simplified: the radioactive trace should be shown in a figure separated from the conjugate (they are two different stories). Also, since the authors showed the trace of the 211-astatine radiolabeled product, they should include the trace of 125-iodininated product too.
Table 2 and Table 4: the authors report small quantities (nanomoles) as moles resulting in numbers with big negative powers. For simplicity and clarity, the authors should use the most suitable fraction of the mole (i.e. nmol, pmol).
Table 1, Table 2, Table 3, Figure 3: How many replicates of the experiments were performed? The RCYs lack ±SD values and the graph in Figure 3 has no error bars. Please indicate the n= and update the values accordingly.
Figure 3: The authors show the kinetics for the TCO-agarose beads only. A graph for the kinetics when magnetic beads were used should be added.
Minor:
1) Some acronyms should be defined e.g. EC, LET
2) MBq/nmol is molar activity (MA), not specific activity.
3) Page 8: it should be “See Table 4”, not “See Table 3”
Reviewer 2 Report
Please see attached file

Reviewer 3 Report
This manuscript brings together two concepts which have been known for 4 decades but which have been studied by only a limited number of groups: At-211 and pretargeting. (Coincidentally, I recall first hearing about both at a conference in the mid 1980s.)
The introduction is well written, with an appropriate level of detail and using mainly recent references. The methods and results are well described.
I have only a few minor comments.
MINOR
At some point the authors should state that I-125 was selected for preclinical work only; for clinical translation I-123 or I-124 would be used.
Specific activity is certainly an issue and the authors describe attempts to achieve higher specific activity. Have the authors estimated the required specific activity for the clinical situation, i.e. a combination of the maximum mass dose of polylysine which could be administered, the required activity for therapeutic effect, and the practical starting activity of At-211?
Effector binding. There should be a little more discussion of how the model relates to the clinical situation. Presumably in vivo the polylysine will be delivered from the blood to the tumour via leaky vessels, then retained for a period related to molecular weight. The authors talk about binding in the first 5 minutes, but also report 24 h values, which I think are unrealistic.
TYPOS ETC
Throughout manuscript: the radioisotope mass numbers should be superscripts
Page 2, Introduction, para 2, line 3: no apostrophe required in “functionalisations”
Page 3 & 4: The title/caption for Table 1 should be on the same page as the table
Page 5, para 2: symbol did not print properly, presumably “micro”
The references are not formatted consistently, e.g. capitalization, number of authors cited
Reference 4: should cite first author rather than working group
Reference 10: Page numbers missing; also name of first author is Gustafsson-Lutz A
Reference 13: citation scrambled, author name Liu G
Reference 18: page numbers missing
Reference 20: page numbers missing